# Resistance to ectromelia virus infection requires cGAS in bone marrow-derived cells which can be bypassed with cGAMP therapy

**Eric B. Wong**[¤], **Brian Montoya, Maria Ferez, Colby Stotesbury, Luis J. Sigal**[*]

Thomas Jefferson University, Department of Microbiology and Immunology, Philadelphia, Pennsylvania, United States of America

¤ Current address: GSK, Innate Immunity Research Unit, Collegeville, Pennsylvania, United States of America

* luis.sigal@jefferson.edu

**Data Availability Statement:** All relevant data are within the manuscript and its Supporting Information files.

## Abstract

Cells sensing infection produce Type I interferons (IFN-I) to stimulate Interferon Stimulated Genes (ISGs) that confer resistance to viruses. During lympho-hematogenous spread of the mouse pathogen ectromelia virus (ECTV), the adaptor STING and the transcription factor IRF7 are required for IFN-I and ISG induction and resistance to ECTV. However, it is unknown which cells sense ECTV and which pathogen recognition receptor (PRR) upstream of STING is required for IFN-I and ISG induction. We found that cyclic-GMP-AMP (cGAMP) synthase (cGAS), a DNA-sensing PRR, is required in bone marrow-derived (BMD) but not in other cells for IFN-I and ISG induction and for resistance to lethal mouse-pox. Also, local administration of cGAMP, the product of cGAS that activates STING, rescues cGAS but not IRF7 or IFN-I receptor deficient mice from mousepox. Thus, sensing of infection by BMD cells via cGAS and IRF7 is critical for resistance to a lethal viral disease in a natural host.

## Author summary

During primary acute systemic viral infections, cells sensing virus through Pathogen Recognition Receptors (PRR) can produce Type I interferons (IFN-I) to induce an anti-viral state that curbs viral spread and protect from viral disease. The dissection of the specific cells, receptors and downstream pathways required for IFN-I production during viral infection *in vivo* is necessary to improve anti-viral therapies. In this study, we demonstrated that the cytosolic PRR cGAS in hematopoietic cells but not in parenchymal cells is required for protection against ectromelia virus, the archetype for viruses that spread through the lympho-hematogenous route. We also show that cGAS deficiency can be bypassed by local administration of cyclic-GMP-AMP (cGAMP) by inducing IFN-I only in the skin and in the presence of virus. Our study provides novel insights into the cGAS signaling pathway and highlights the potential of cGAMP as an efficient anti-viral treatment.

**Funding:** This work was supported by National Institute of Allergy and Infectious Diseases (NIAID, https://www.niaid.nih.gov) of the National Institutes of Health (NIH) by grants R01AI065544 and R01AI110457 to LJS and F32AI129352 to EBW. Research reported in this publication utilized the Flow Cytometry and Laboratory Animal facilities at Sidney Kimmel Cancer Center at Thomas Jefferson University and was supported by the National Cancer Institute (NCI, https://www.cancer.gov) of the NIH under Award Number P30CA056036. The funders had no role in study design, data collection and analysis, decision to publish, or preparation of the manuscript.

## Introduction

Numerous viruses relevant to human and animal health utilize a lympho-hematogenous route of dissemination whereby they penetrate their hosts though disruptions of epithelial surfaces such as the skin, spread to the draining lymph node (dLN) *via* afferent lymphatics, and become systemic by disseminating to the blood through efferent lymphatics [1]. However, our understanding of how the host innate immune system mechanistically induces protective resistance to the virus during lympho-hematogenous dissemination is incomplete. Ectromelia virus (ECTV), a member of the Orthopoxvirus genus of large, closely-related DNA viruses and the causative agent of mousepox (the mouse homolog of human smallpox) is the archetype used to study lympho-hematogenous dissemination [1–4].

Type I interferons (IFN-I), which in the mouse, include one IFN-β and 14 IFN-α, are produced rapidly following viral infection. IFNs induce a Janus kinase signal transducer and activator of transcription-mediated (JAK-STAT) pathway that leads to the transcriptional regulation of several hundred interferon stimulated genes (ISGs) that are collectively involved in the coordinated effort to resist and control viral spread and pathogenicity [5–7]. Two lines of evidence demonstrate the importance of IFN-I in resistance to mousepox. First, C57BL/6(B6) mice are resistant to mousepox, but B6 mice deficient in the IFN-I receptor (IFNAR) subunit IFNAR1 have very high levels of viral replication, dissemination and succumb to mousepox [8–10]. Second, ECTV encodes EVM166, a IFN-I decoy receptor, that is critical for ECTV virulence in susceptible mice but is still pathogenic in IFNAR1 deficient mice [9]. Notably, EVM166 inactivates mouse IFNα but not IFNβ suggesting that only IFNα is critical for resistance to mousepox [9].

The production of IFN-I, and also of many inflammatory cytokines and chemokines, requires sensing of infection through pathogen recognition receptors (PRR) [11–14]. PRRs are a class of germline-encoded receptors that recognize pathogen-associated molecular patterns (PAMP), typically conserved molecular motifs with repetitive structures such as lipopolysaccharide, RNA and DNA [11]. PAMPs in the cell microenvironment are surveyed by membrane bound PRRs that are either at the cell surface or on endocytic compartments, such as Toll-like receptor 9 (TLR9) which recognizes viral CpG DNA. The cytosolic TIR domain of TLR9 and most other TLRs directly interact with the adapter Myeloid differentiation primary response 88 (MyD88) which becomes phosphorylated upon PAMP sensing. This activates a cascade of signaling molecules that culminates in the activation of transcription factors such as IRF3, IRF7, and NF-κB that induce the transcription of IFN-I and/or proinflammatory cytokines and chemokines. PRRs in in the cytoplasm such as RIG-I like helicases (RLH) recognize viral RNA, while Interferon-activated gene 204 (Ifi204, IFI16 in humans), and cyclic-GMP-AMP (cGAMP) synthase (cGAS) recognize viral DNA [12–15]. cGAS synthesizes the messenger cGAMP which bind Stimulator of Interferon Genes (STING) to induce IRF3, IRF7, NF-κB or other transcription factors to stimulate the transcription of IFN-I and chemokines and cytokines [12]. Ifi204/IFI16 also works through STING but the specific mechanism is not understood.

We have previously demonstrated cooperation between sensing pathways in response to ECTV, in which MHC-II$^{hi}$ dendritic cells (DCs) from the skin migrate to the dLN and produce the chemokines CCL2 and CCL7 in a TLR9-MyD88-IRF7 dependent manner to recruit inflammatory monocytes (iMO) to the dLN [16,17]. We also showed that in the dLN iMOs become infected and use STING-IRF7 and STING-NF-κB to respectively induce IFNα and IFNβ [16]. We also showed that during lympho-hematogenous ECTV spread, iMOs and not plasmacytoid DCs (pDCs) are the main producers of IFN-I. Yet, the PRR(s) responsible for IFN-I induction upstream of STING remained unidentified. Very recently, Cheng et al.

showed that cGAS upstream of STING is important for IFNβ production and resistance to mousepox with high ECTV doses [18] but its role in the induction of the more critical IFNα or ISGs and the *in vivo* cellular requirements were not investigated. Here we confirm that cGAS is necessary for resistance to mousepox, in our case to relatively low viral doses, and that it is important for efficient IFNβ expression. We also demonstrate that cGAS is important for optimal expression of more relevant IFNα and also ISGs. Furthermore, we demonstrate that resistance to mousepox necessitates cGAS in bone marrow-derived (BMD) cells but not in parenchymal cells. We further show that cGAMP administration at the site of infection prevents lethal mousepox in cGAS deficient mice by curbing virus spread through the induction of IFN-I and ISGs. Mechanistically, cGAMP rescue requires the transcription factor IRF7 and the IFN-I receptor IFNAR1. Together, these findings demonstrate that early sensing of virus by BMD cells *via* cGAS is critical to control a highly pathogenic DNA viral infection, and highlights the importance that curbing virus spread from the initial site of infection.

## Methods

### Mice

All mice used in experiments were 6–12 weeks old. C57BL/6 (B6) mice were purchased from Charles River. $Cgas^{-/-}$ mice were a gift from Zhijian Chen (UT Southwestern Medical Center, Dallas TX). $Tmem173^{gt}$ (C57BL/6J-$Tmem173^{gt}$/J) were originally purchased from Jackson Laboratories (Bar Harbor, ME). $Irf7^{-/-}$ (B6.129P2-Irf7$^{tm1Ttg}$/TtgRbrc) were from Riken Bioresource Center (Tsukuba, Japan). $Ifnar1^{-/-}$ mice backcrossed to B6 were a gift from Dr. Thomas Moran (Mount Sinai School of Medicine, New York, NY). $Ifi204^{-/-}$ mice were generated from frozen germplasm from the Knockout Mouse Project (KOMP) Repository (UC Davis, Davis CA). Colonies were bred at Thomas Jefferson University under specific pathogen free conditions.

### Viruses and infection

Virus stocks, including ECTV-Moscow strain (WT) (ATCC VR-1374), ECTV-EGFP [19] and ECTV-dsRED [20] were propagated in tissue culture as previously described [9]. Mice were infected with 3,000 plaque forming units (PFUs) ECTV-WT, ECTV-EGFP or ECTV-dsRED as indicated. For the determination of survival, mice were monitored daily and to avoid unnecessary suffering, mice were euthanized and counted as dead when imminent death was certain as determined by lack of activity and unresponsiveness to touch. Euthanasia was according to the 2013 edition of the AVMA Guideline for the Euthanasia of Animals. For virus titers, the entire spleen or portions of the liver were homogenized in RPMI using a Tissue Lyser (QIAGEN). Virus titers were determined on BSC-1 cells as described before [9].

### Production of bone marrow chimeric mice

Bone marrow chimeras were prepared as previously described [21]. Briefly, 6- to 8-week-old mice were irradiated with 900 rads using a GammaCell 40 apparatus (Nordion Inc.). Irradiated mice were reconstituted intravenously with an inoculation of 2-5x10$^6$ bone marrow cells from different donors. Chimeras were administered with acidified water and rested for 8 weeks after reconstitution.

### cGAMP administration

To investigate the effect of exogenous cGAMP during infection, 10 ug of 2'3' cGAMP (Invivogen) or control PBS was injected into the footpad 4 hours prior to infected with ECTV.

## Flow cytometry

Flow cytometry was performed as previously described [9]. The following Abs were used: BV785-CD3ε (Clone 145-2C11), BV711-CD8α (Clone 53–6.7), APC/Cy7-CD11b (Clone M1/70), BV-421-Gr-1 (Clone RB6-8C5), PE/Cy7-CD11c (Clone N418), FITC-CD45 (Clone I3/2.3), PerCP/Cy5.5-IA$^b$ (Clone AF6-120.1), BV605-NK1.1 (PK136) were from Biolegend. BUV395-CD4 (Clone GK1.5) and BUV395-CD19 (Clone 1D3) were from BD Biosciences. To obtain single-cell suspensions, LNs were first incubated in Liberase TM (1.67 Wünsch units/mL) (Sigma) in PBS with 25 mM HEPES for 30 min at 37°C before adding PBS with 25 mM HEPES + 10% FBS to halt the digestion process, followed by mechanical disruption of the tissue through a 70-μm filter. Cells were washed once with FACS buffer (PBS with 1% BSA + 0.1% sodium azide) before surface staining. For analysis, samples were acquired using either a BD LSR II flow cytometer or a BD Fortessa flow cytometer (BD Biosciences), and data were analyzed with FlowJo software (TreeStar). For cell sorting, samples were acquired with a BD FACS Aria III sorter (BD Bioscience).

## RNA preparation and RT-qPCR

Total RNA from LNs or sorted cells ($10^4$–$10^5$ cells) was obtained with the RNeasy Mini Kit (QIAGEN) as previously described [10,16]. First-strand cDNA was synthesized with High Capacity cDNA Reverse Transcription Kit (Life Technologies). qPCR was performed as before [10,16] using primers as follows: *Gapdh*: tgtccgtcgtggatctgac and cctgcttcaccaccttcttg, *Evm003*: tctgtcctttaacagcatagatgtagа and tgttaactcggaagttgatatggta, *Evm036*: tgccagttagcactgcgtat and aggtgttctggagaatcaaaga, *Ifng*: gcaaaaggatggtgacatga and ttcaagacttcaaagagtctgaggta, *Ccl2*: catccacgtgttggctca and gatcatcttgctggtgaatgagt, *Ccl7*: ttgacatagcagcatgtggat and ttctgtgcctgctgctcata, *Cxcl9*: cttttcctcttgggcatcat and gcatcgtgcattccttatca, *Il6*: tctaattcatatcttcaaccaagagg and tggtccttagccactccttc, *Tnf*: tcttctcattcctgcttgtgg and ggtctgggccatagaactga, *Mx1*: gatccgacttcacttccagatgg and catctcagtgtagtcaaccc, *Il12*: gactccaggggacaggcta and ccaggagatggttagcttctg, *Ifit3*: tgaactgctcagcccaca and tcccggttgacctcactc, *Isg15*: agtcgacccagtctctgactct and ccccagcatcttcacctttta, *Irf7*: tgtagtgtggtgacccttgc and cttcagcactttcttccgaga, *Ifnb1*: catttccgaatgttcgtcct and cacagccctctccatcaacta, *Ifna4*: gtcttttgatgtgaagaggttcaa and tcaagccatccttgtgctaa, *Ifna-non4*: aagctgtgtgatgcaacaggt and ggaacacagtgatcctgtgg.

## Statistics

Data were analyzed with Prism 6 software (GraphPad Software). For survival we used the Log-rank (Mantel-Cox). For other experiments ANOVA with Tukey correction for multiple comparisons or Student's t test were used as applicable. In all figures, *p<0.05, **p<0.01, ***p<0.001, ****p<0.001.

## Ethics statement

All the procedures involving mice were carried out in strict accordance with the recommendations in the Eight Edition of the Guide for the Care and Use of Laboratory Animals of the National Research Council of the National Academies. All experiments were approved by Thomas Jefferson University's Institutional Animal Care and Use Committee under protocol number 01727 "Immune Control of Viral Infections".

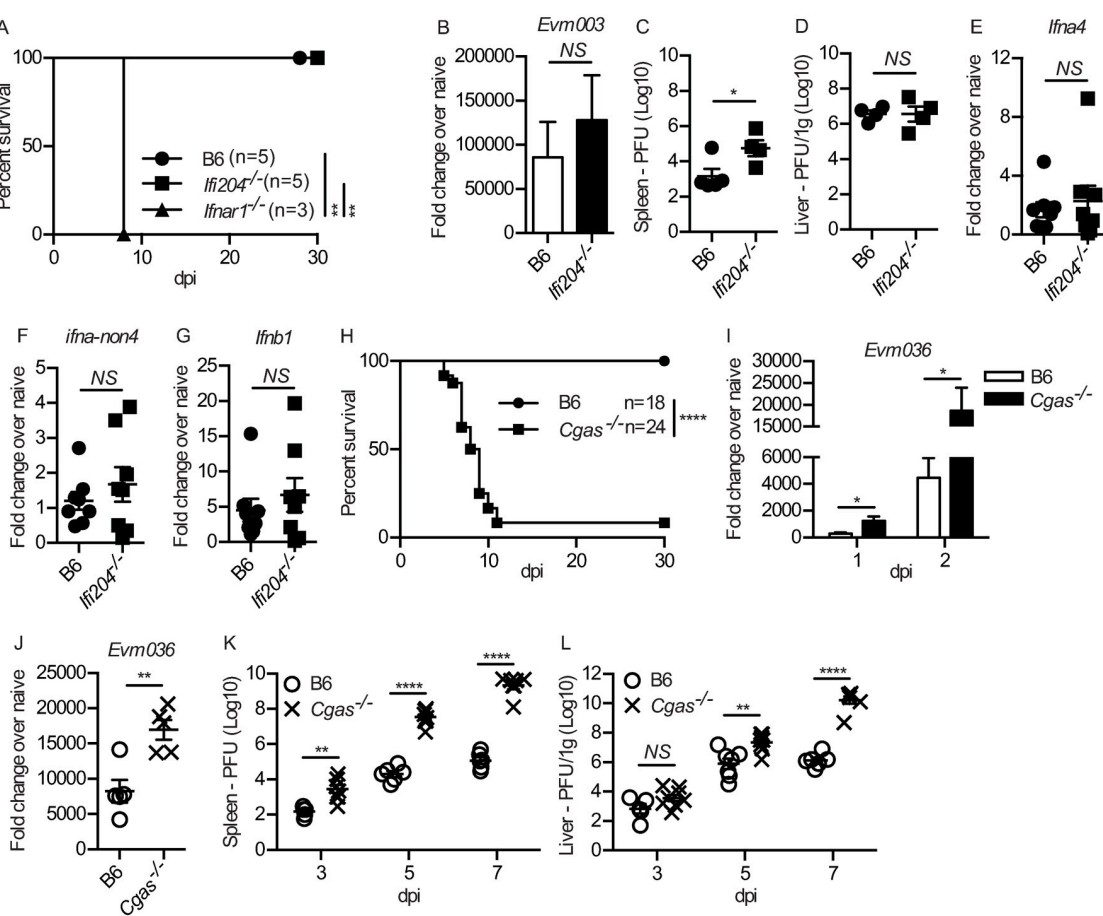

**Fig 1. cGAS, but not IFI204, is required for optimal resistance to ECTV infection.** **(A)** Survival of the indicated mice. **(B)** Expression of *Evm036* in the dLN at 2 dpi of the indicated mice as determined by RT-qPCR. Data are displayed as mean ± SEM from 5 mice per group in one experiment, which is representative of two similar experiments. **(C-D)** Virus loads in the spleen (C) and liver (D) of the indicated mice at 7 dpi as determined by plaque assay. Data are displayed as the mean ± SEM of 4–5 mice per group. **(E-G)** Expression of IFN-I in the dLN at 2.5 dpi of the indicated mice as determined by RT-qPCR. Data are displayed as mean ± SEM from 5 mice per group in one experiment, which is representative of three similar experiments. **(H)** Survival of the indicated mice. **(I-J)** Expression of *Evm036* in the skin (1 or 2 dpi) (I) and dLN (2 dpi) (J) of the indicated mice as determined by RT-qPCR. Data are displayed as mean ± SEM from 5 mice per group in one experiment, which is representative of three similar experiments. **(K-L)** Virus loads in the spleen (K) and liver (L) of the indicated mice at 3, 5 or 7 dpi as determined by plaque assay. Data are displayed as the mean ± SEM of 5–6 mice per group. For all, *p<0.05, **p<0.01, ***p<0.001, ****p<0.0001.

## Results

### cGAS, but not IFI204, is not required for resistance to ECTV infection

Our previous work established the important role of STING, IRF7 and NF-κB in IFN-I induction during ECTV lympho-hematogenous dissemination [16]. We had also explored a possible role for the DNA-dependent activator of IFN-regulator factors (DAI/Zbp1) as the PRR upstream of STING, but we found it was not required for resistance to ECTV or IFN-I induction [16]. Thus, we set to identify the PRR upstream of STING. *Ifi204* is one of the few genes that mapped as a possible candidate to the still unidentified Resistance to Mousepox 4 (Rmp-4) gene [22]. Notably, murine IFI204 and its human ortholog IFI16 are members of the PRR AIM2-like receptor family and have been shown to signal through STING [23–26]. This made *Ifi204* an intriguing candidate as an ECTV PRR. However, all *Ifi204*⁻/⁻ mice survived ECTV infection (Fig 1A) and did not exhibit significant differences with WT B6 mice in viral

replication in the dLN at 2.5 dpi, measured by reverse transcriptase quantitative polymerase chain reaction (RT-qPCR) as mRNA for the ECTV gene *Evm036* (Fig 1B). Also, while *Ifi204*$^{-/-}$ mice had a slight but significantly higher virus titers in the spleen at 7 dpi, there were no significant differences in viral burden in the liver compared to B6 controls (Fig 1C and 1D). Furthermore, *Ifi204*$^{-/-}$ mice had similar levels of mRNA for *Ifna-non4* (detected by RT-qPCR with a single primer pair as mRNA for all IFN-α except α4 [27] and *Ifnb1* (encoding the single IFN-β) expression in the dLN compared to B6 mice (Fig 1E and 1G). Thus, IFI204 is not a significant cytosolic PRR for IFN-I expression in response to ECTV infection or for ECTV control.

We investigated a possible role for cGAS, because it had been shown important for the control of the DNA virus herpes simplex-1 [28]. We found that most B6 mice deficient in cGAS (*Cgas*$^{-/-}$) succumbed to mousepox (Fig 1H). Consistently, viral loads, measured by *Evm036* mRNA transcripts in the skin of the footpad at 1 and 2 dpi (Fig 1I) and in the dLN at 2 dpi (Fig 1J), were significantly higher in *Cgas*$^{-/-}$ mice compared to B6 controls. Furthermore, the unrestricted viral replication permitted swift dissemination to peripheral organs, as *Cgas*$^{-/-}$ mice had significantly higher viral titers in the spleen at 3, 5 and 7 dpi (Fig 1K) and liver at 5 and 7 dpi (Fig 1L). Thus, cGAS is necessary for ECTV control and resistance to lethal mousepox.

## *Cgas*$^{-/-}$ mice have a defective IFN-I and ISG response in the dLN after infection

To determine whether cGAS deficiency affects IFN, chemokines, cytokines and ISG gene transcription in the early innate immune response to ECTV during lympho-hematogenous spread, we isolated mRNA from the dLN of infected B6 and *Cgas*$^{-/-}$ mice at 2.5 dpi and performed RT-qPCR to measure relative fold-change in gene expression over each group's contralateral ndLN. We found that cGAS deficiency resulted in significantly decreased, but not absent, transcription of the IFN-I genes *Ifna4* (Fig 2A), *Ifna-non4* (Fig 2B) and *Ifnb1* (Fig 2C), which at this time point are mostly produced by infected iMOs [16]. However, cGAS deficiency did not affect IFN-γ gene expression (Fig 2D), which is produced mostly by NK cells [17,19,29]. Different to WT mice, *Cgas*$^{-/-}$ mice did not upregulate expression of the ISGs *Ifit3* (Fig 2E), *Irf7* (Fig 2F) *Isg15* (Fig 2G), and *Mx1* (Fig 2H) suggesting that the induction of ISGs by IFN-I and IFN-γ may be non-overlapping *in vivo*. To ensure that the significant decrease in IFN-I and ISG expression is inherently due to cGAS deficiency and not a challenge-specific phenomenon, we inoculated B6 and *Cgas*$^{-/}$ mice with either ECTV or TLR9 agonist CpG and found that CpG inoculation in the footpad does not induce IFN-I or ISG expression at 2 d post challenge (*Il12* expression was used as a positive control and was found to be upregulated in both B6 and *Cgas*$^{-/-}$ mice) (S1 Fig). cGAS deficiency did not affect the transcription of the chemokine genes *Ccl2* (Fig 2I) and *Ccl7* (Fig 2J) that are important for iMO recruitment [16], or *Cxcl9* (Fig 2K), which is important for NK cell recruitment and dependent on IFN-γ during ECTV infection [17]. Also, cGAS deficiency did not affect the transcription of the pro-inflammatory cytokine genes *Il12* (Fig 2L), *Il6* (Fig 2M) and *Tnf* (Fig 2N). Together, these data suggest that the accelerated viral dissemination and increased susceptibility of *Cgas*$^{-/-}$ mice to mousepox is likely due to significantly reduced global IFN-I production in the dLN and consequently, a dampened ISG response.

## cGAS expression in hematopoietic-derived cells is required for optimal IFN-I and ISG expression in the dLN

Next, we made bone marrow chimeras to investigate whether cGAS expression in the hematopoietic compartment is required for an optimal IFN-I response and resistance to ECTV. The degree of donor BM reconstitution in the irradiated hosts was successful as there were

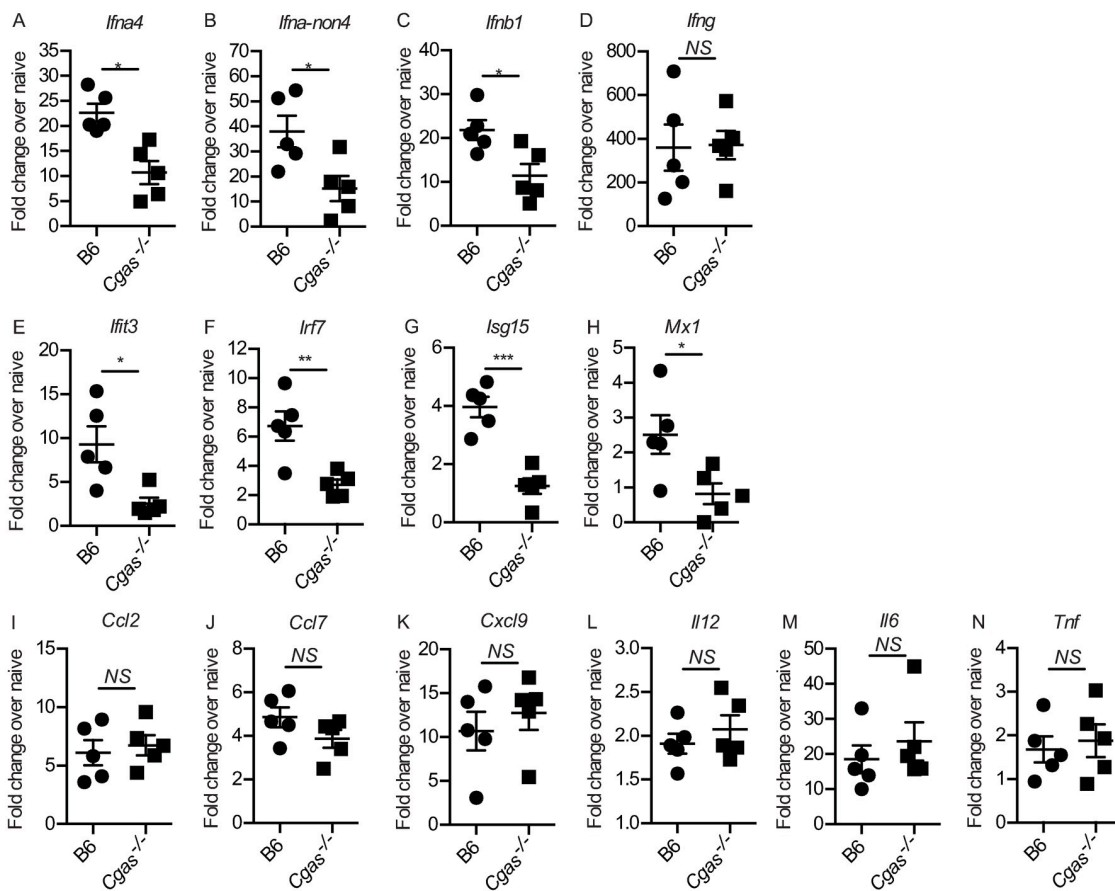

**Fig 2. *Cgas<sup>-/-</sup>* mice have a defective IFN-I and ISG response in the dLN after infection. (A-N)** Expression of IFN-Is (A-C), IFN-γ (D), ISGs (E-H) and proinflammatory chemokines/cytokines (I-N) in the dLN at 2.5 dpi of the indicated mice as determined by RT-qPCR. Data are displayed as mean ± SEM from 5 mice per group in one experiment, which is representative of three similar experiments. For all, *p<0.05, **p<0.01, ***p<0.001.

approximately equivalent cellularity and total numbers of different immune cell subsets in the naïve contralateral LNs in all of the different groups (S2 Fig). We found that *Cgas<sup>-/-</sup>→Cgas<sup>-/-</sup>* and *Cgas<sup>-/-</sup>→*B6 chimeras succumbed to mousepox, while B6→ *Cgas<sup>-/-</sup>* and B6→B6 chimeras survived (Fig 3A). In agreement, compared to B6→B6 and B6→*Cgas<sup>-/-</sup>* mice, *Cgas<sup>-/-</sup>→Cgas<sup>-/-</sup>* and *Cgas<sup>-/-</sup>→*B6 mice had significantly higher virus transcripts in the dLN at 2.5 dpi (Fig 3B) and higher infectious virus in the spleen (Fig 3C) and liver (Fig 3D) at 7 dpi. This indicates that ECTV control and resistance to mousepox requires cGAS expression in the hematopoietic compartment and not in the parenchyma. Accordingly, *Cgas<sup>-/-</sup>→Cgas<sup>-/-</sup>* and *Cgas<sup>-/-</sup>→*B6 chimeras had significantly lower IFN-I expression in the dLN (Fig 3E–3G) but not *Ifng* (Fig 3H), *Cxcl9* (Fig 3I), *Il6* (Fig 3J), *Il12* (Fig 3K) or *Tnf* (Fig 3L). Finally, cGAS expression in the hematopoietic compartment was required for the expression of ISGs as *Cgas<sup>-/-</sup>→Cgas<sup>-/-</sup>* and *Cgas<sup>-/-</sup>→*B6 chimeras had significantly reduced levels of ISGs in the dLN at 2.5 dpi (Fig 3M–3P).

## *Cgas<sup>-/-</sup>* mice have an intrinsic defect in IFN-I expression and accumulation of MHC-II<sup>hi</sup> DCs, iMOs and NK cells

We have shown that both MHC-II<sup>hi</sup> DCs and iMOs are important sources of IFN-I in the dLN after ECTV infection; MHC-II<sup>hi</sup> DCs provide the early source of IFN-I in the dLN

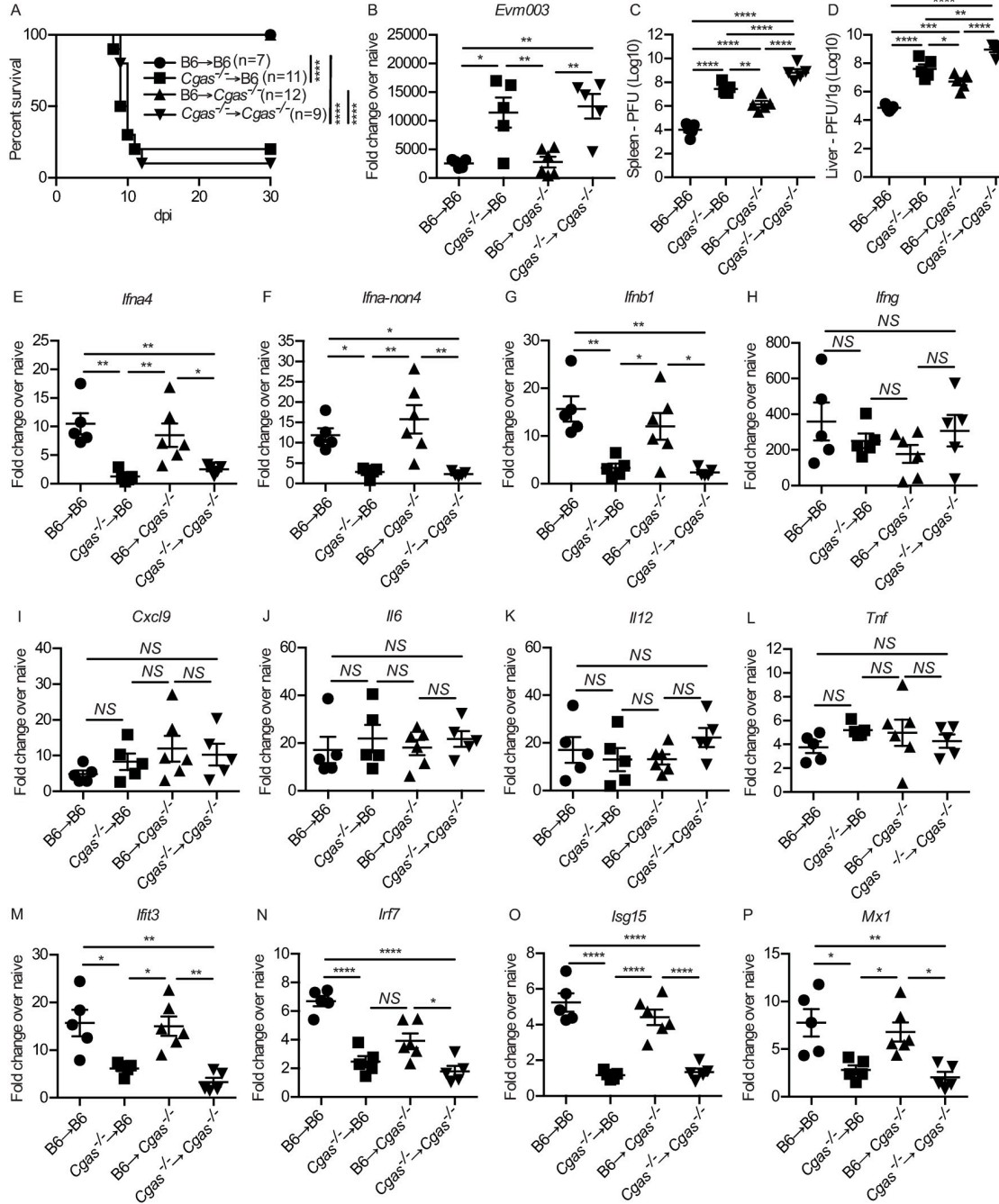

**Fig 3. cGAS expression in hematopoietic-derived cells is required for optimal IFN-I and ISG expression in the dLN. (A)** Survival of the indicated mice **(B)** Expression of *Evm003* in the dLN of the indicated mice at 2 dpi as determined by RT-qPCR. Data are displayed as mean ± SEM from 5 mice per group in one experiment, which is representative of two similar experiments. **(C-D)** Virus loads in the spleen (C) and liver (D) of the indicated mice at 7 dpi as determined by plaque assay. Data are displayed as the mean ± SEM of 5 mice per group. **(E-P)** As in (B) measuring *Ifna4* (E), *Ifna-non4* (F), *Ifnb1* (G), *Ifng* (H), *Cxcl9* (I), *Il6* (J), *Il12* (K) *Tnf* (L), *Ifit3* (M), *Irf7* (N) *Isg15* (O) and *Mx1* (P). Data are displayed as mean ± SEM from 5 mice per group in one experiment, which is representative of two similar experiments. For all, *p<0.05, **p<0.01, ***p<0.001, ****p<0.0001.

([60]) and are the principal cells that recruit iMOs, before infected iMOs take over as the major source of IFN-I in the dLN at later time points [16]. To determine if the increased viral dissemination in *Cgas*[-/-] mice was due to altered recruitment or functionality of MHC-II[hi] DCs, we infected B6 and *Cgas*[-/-] mice and analyzed the kinetics of MHC-II[hi] DC recruitment to the dLN and found that the frequency of MHC-II[hi] DCs did not change at 1–3 dpi (Fig 4A and 4B) however, there was a significant reduction in total numbers of MHC-II[hi] DCs in the dLN at both 2 and 3 dpi in *Cgas*[-/-] mice (Fig 4C). We further confirmed that cGAS deficiency did not inherently alter the ability of MHC-II[hi] DCs to migrate to the dLN through the generation of 50:50 mixed B6+*Cgas*[-/-] → B6 bone marrow chimeras and found equivalent frequencies and ratios of MHC-II[hi] DCs in the dLN at 2 dpi (Fig 4D and 4E). *Cgas*[-/-] MHC-II[hi] DCs appeared functionally intact, at least with respect to expression of intracellular CXCL9 (S3A Fig), and surface expression of IFNAR1 (S3B Fig), costimulatory molecules CD40, CD80 and CD86 (S3C–S3E Fig), and the NKG2D ligands Rae1 and MULT1 which are necessary to active NK cells in the dLN [17,19] (S3F and S3G Fig). However, at 2.5 dpi with ECTV-dsRED, a significantly higher proportion of MHC-II[hi] DCs were dsRED[+] in the dLN of *Cgas*[-/-] mice compared to B6 indicating that *Cgas*[-/-] MHC-II[hi] DCs are more susceptible to infection (Fig 4F). When we sorted dsRED[+] infected and dsRED[-] uninfected MHC-II[hi] DCs from B6 and *Cgas*[-/-] mice, we found that *Cgas*[-/-] MHC-II[hi] DCs had significantly higher levels of viral gene mRNA at 2 dpi, suggesting that cGAS also has a cell-intrinsic role in anti-viral protection (Fig 4G). Moreover, ECTV-dsRED[-] MHC-II[hi] DCs from *Cgas*[-/-] mice had relatively high levels of viral transcription, likely indicating early infection, before dsRED becomes fluorescent (Fig 4G). *Cgas*[-/-] MHC-II[hi] DCs did not upregulate *Ifna4* (Fig 4H), *Ifna-non4* (S3H Fig) and *Ifnb1* (S3I Fig) and the ISGs *Ifit3* (Fig 4I), *Irf7* (S3J Fig), *Isg15* (S3K Fig) and *Mx1* (S3L Fig).

Next, we analyzed whether cGAS deficiency altered iMO accumulation in the dLN. The frequencies of iMO in *Cgas*[-/-] and B6 mice were similar at 1–3 dpi (Fig 4J–4K). However, *Cgas*[-/-] mice had a significant reduction in the total number of iMO in the dLN at 2 and 3 dpi (Fig 4L). Yet, *Cgas*[-/-] iMOs did not have an intrinsic defect in recruitment to the dLN as determined in 50:50 mixed B6-*Cgas*[-/-] bone marrow chimeras (Fig 4M and 4N). *Cgas*[-/-] iMOs were more susceptible to infection (Fig 4O) and did not upregulate IFN-I (*Ifna4* (Fig 4P), *Ifna-non4* (S3M Fig) and *Ifnb1* (Fig 1N) or the ISGs (*Ifit3* (Fig 4Q), *Irf7* (S3O Fig), *Isg15* (S3P Fig) and *Mx1* (S3Q Fig) but did not differ from B6 iMOs in the frequencies that were CXCL9[+] or TNF-α[+] (S3R and S3S Fig).

Optimal resistance to ECTV requires a robust NK cell response [19,29]. Our previous work demonstrated that MHC-II[hi] DCs and iMOs play a critical role in the induction and recruitment of circulating NK cells to the dLN [17]. Therefore, we next determined whether *Cgas*[-/-] mice have an altered NK cell response. As with MHC-II[hi] DCs and iMOs, *Cgas*[-/-] mice had similar frequencies of NK cells in the dLN at 1–3 dpi (Fig 4R and 4S), but significant reduction in total numbers of NK cells at 2 and 3 dpi (Fig 4T). *Cgas*[-/-] NK cells were significantly more permissive to infection (Fig 4U), however they were functionally intact, as *Cgas*[-/-] mice had no significant differences in frequency of Granzyme B[+], IFN-γ[+] and TNF-α[+] NK cells in the dLN compared to B6 controls (S3T–S3V Fig).

We hypothesized that the decreased IFN-I expression and increased infection rate of MHC-II[hi] DCs and iMOs in the dLN of *Cgas*[-/-] mice could increase viral-induced cell death without affecting frequencies of MHC-II[hi] DCs and iMOs. In agreement, at 3 dpi, there was a significant higher frequency of Zombie[+] (dead) cells in the dLNs of *Cgas*[-/-] than of B6 mice (Fig 4V).

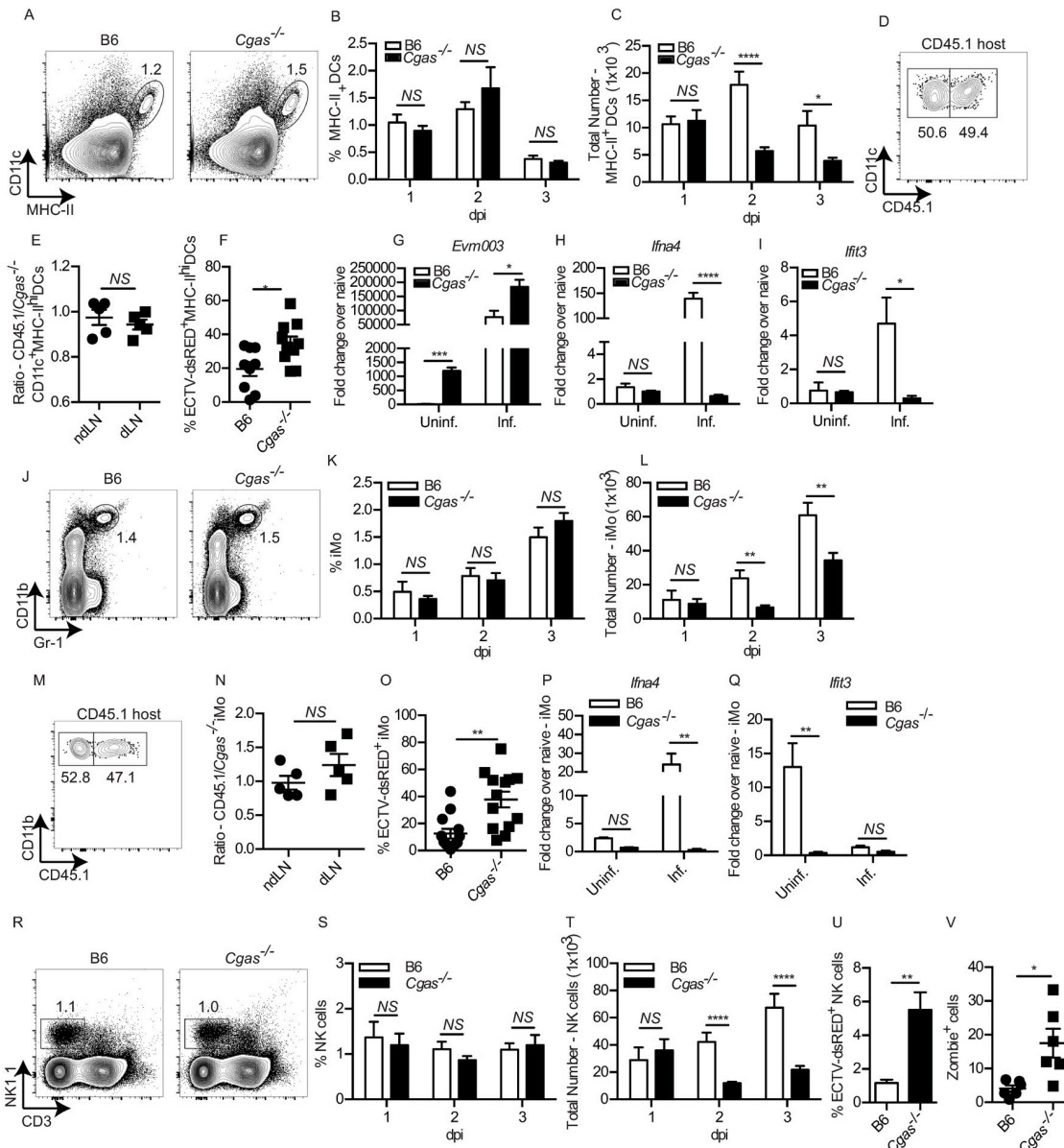

**Fig 4. *Cgas*[-/-] mice have an intrinsic defect in IFN-I expression and accumulation of MHC-II[hi] DCs, iMOs and NK cells. (A)** Representative flow cytometry plots of MHC-II[hi] DC frequency in the dLN at 2 dpi. **(B-C)** Calculated frequency (B) and total numbers (C) of MHC-II[hi] DCs in the dLNs of the indicated mice at the indicated dpi. Data are displayed as the mean ± SEM of 12–15 mice per group combined from three independent experiments. **(D)** Representative flow cytometry plots of CD45.1 expression on MHC-II[hi] DCs from bone marrow chimeras reconstituted with a 50:50 mix of B6 and *Cgas*[-/-] bone marrow cells. **(E)** Calculated ratio of CD45.1/*Cgas*[-/-] MHC-II[hi] DCs from the ndLNs and dLNs of 50:50 mixed bone marrow chimeras at 2 dpi. **(F)** Frequency of ECTV-dsRED[+]MHC-II[hi] DCs in the dLN of the indicated mice at 2 dpi. **(G-I)** Expression of mRNA for the indicated genes: *Evm003* (G), *Ifna4* (H) and *Ifit3* (I) from sorted uninfected and infected MHC-II[hi] DCs from the dLN of the indicated mice at 2 dpi. Data are displayed as mean ± SEM of pooled cells from 6–8 mice per group in one experiment, which is representative of two similar experiments. P values were calculated based on three technical replicates. **(J)** Representative flow cytometry plots of CD11b[+]Gr-1[int] iMO in the dLN of the indicated mice at 2.5 dpi. **(K-L)** As in (B-C) but for iMO. **(M-N)** As in D-E but for iMO. **(O)** Frequency of ECTV-dsRED[+] iMO in the dLN of the indicated mice at 2.5 dpi. **(P-Q)** As in G-K but for sorted infected and uninfected iMO from the indicated mice at 2 dpi. **(R)** Representative flow cytometry plots of NK1.1[+]CD3[-] NK cells. **(S-T)** As in D-E but for NK cells. **(U)** Frequency of ECTV-dsRED[+] NK cells in the dLN of the indicated mice at 2.5 dpi. **(T)** Calculated frequency of Zombie[+] cells in the dLN at 3 dpi from the indicated mice. For all, *p<0.05, **p<0.01, ***p<0.001, ****p<0.0001.

## Administration of cGAMP to *Cgas*<sup>-/-</sup> mice restores IFN-I expression and resistance to ECTV infection

Binding of DNA to cGAS catalyzes the production of cGAMP which functions as a second messenger that activates STING [30,31]. Therefore, we tested whether exogenous administration of cGAMP restores resistance to mousepox to *Cgas*$^{-/-}$ mice. All *Cgas*$^{-/-}$ mice that received cGAMP in the footpad prior to infection (cGAMP+ECTV) survived (Fig 5A) with a concomitant significant reduction in viral replication in the dLN at 2 dpi (Fig 5B). To determine whether the decrease in viral replication was due to cGAMP-mediated IFN-I induction in the dLN, we measured IFN-I transcripts in the dLN at 2 dpi, however, there was a drastic reduction in *Ifna4* (Fig 5C) *Ifna-non4* (Fig 5D) and *Ifnb1* (Fig 5E).). Although cGAMP administration increased the recruitment of MHC-II$^{hi}$ DCs to the dLN of both B6 and *Cgas*$^{-/-}$ mice (Fig 5F), there was a significant decrease in iMOs (Fig 5G), which accounted for the decrease in IFN-I. These results suggested that cGAMP decreases viral replication in the skin of the footpad and curbs (but does not prevent) spread to the dLN. In agreement, at 2 dpi, there was significantly lower viral replication in the skin of the footpad in cGAMP+ECTV mice (Fig 5H). Moreover, cGAMP+ECTV induced a strong increase in *Ifna4*, *Ifna-non4*, and *Ifnb1* expression in the skin of the footpad of B6 and *Cgas*$^{-/-}$ mice (Fig 5I–5K). Furthermore, cGAMP+ ECTV restored the expression of the ISGs *Ifit3*, *Irf7*, *Isg15* and *Mx1* in the skin (Fig 5L and 5M), therefore, potentially contributing to diminishing viral replication in the skin with subsequent decrease in dissemination to the dLN.

## cGAMP-mediated resistance to ECTV is mostly dependent on IRF7

It is well-established that pathogen-recognition by cGAS catalyzes the formation of cGAMP which activates STING [28,32]. While STING has been demonstrated to activate not only IRF3 but also IRF7 [33,34], IRF3 has been overwhelmingly utilized to describe the canonical pathway of cGAS-STING signaling. However, we have previously demonstrated that IRF7, but not IRF3, is required for T1-IFN expression in the dLN and resistance to ECTV [16]. Therefore, we investigated whether IRF7 is required for the anti-ECTV effects of cGAMP. We found that cGAMP administration to *Irf7*$^{-/-}$ mice did not decrease ECTV gene expression in the skin (Fig 6A) or dLN at 2 dpi (Fig 6B). Also, ECTV-infected *Irf7*$^{-/-}$ mice receiving cGAMP and ECTV upregulated transcription of *Ifna4* (Fig 6C and 6F) and *Ifnb1* (Fig 6D and 6G) but not of *Ifna-non4* (Fig 6E and 6H) in the skin and dLN. IRF7 deficiency did not alter expression levels of proinflammatory cytokines and chemokines in the skin and dLN after ECTV or cGAMP +ECTV (S4A–S4H Fig) indicating that the main role of IRF7 during ECTV infection is in IFN-I but not in pro-inflammatory cytokine induction. Furthermore, despite the *Ifna4* and *Ifnb1* increase, cGAMP and cGAMP+ECTV did not upregulate ISGs in the skin (Fig 6I–6L) of *Irf7*$^{-/-}$ mice. Also, the seemingly increased survival of *Irf7*$^{-/-}$ mice treated with cGAMP in survival as compared to untreated controls, did not reach statistical significance (Fig 6M).

## cGAMP-mediated ISG induction is dependent on IFNAR1

The induction of ISGs requires IFN-I binding and signaling through IFNAR1 to induce an anti-viral state [6,7]. After ECTV infection, *Ifnar1*$^{-/-}$ mice had significantly higher levels of ECTV gene transcription than B6 controls in the skin of the footpad and in the dLN at 2 dpi whether receiving cGAMP or not (Fig 7A and 7B). Furthermore, while IFN-I expression in the skin (Fig 7C–7E) and dLN (Fig 7F–7H) was unaffected by the absence of IFNAR1, *Ifnar1*$^{-/-}$ mice were unable to upregulate ISGs (Fig 7I–7L). IFNAR1 deficiency also did not alter proinflammatory cytokine and chemokine expression in the skin (S5A–S5D Fig) or dLN

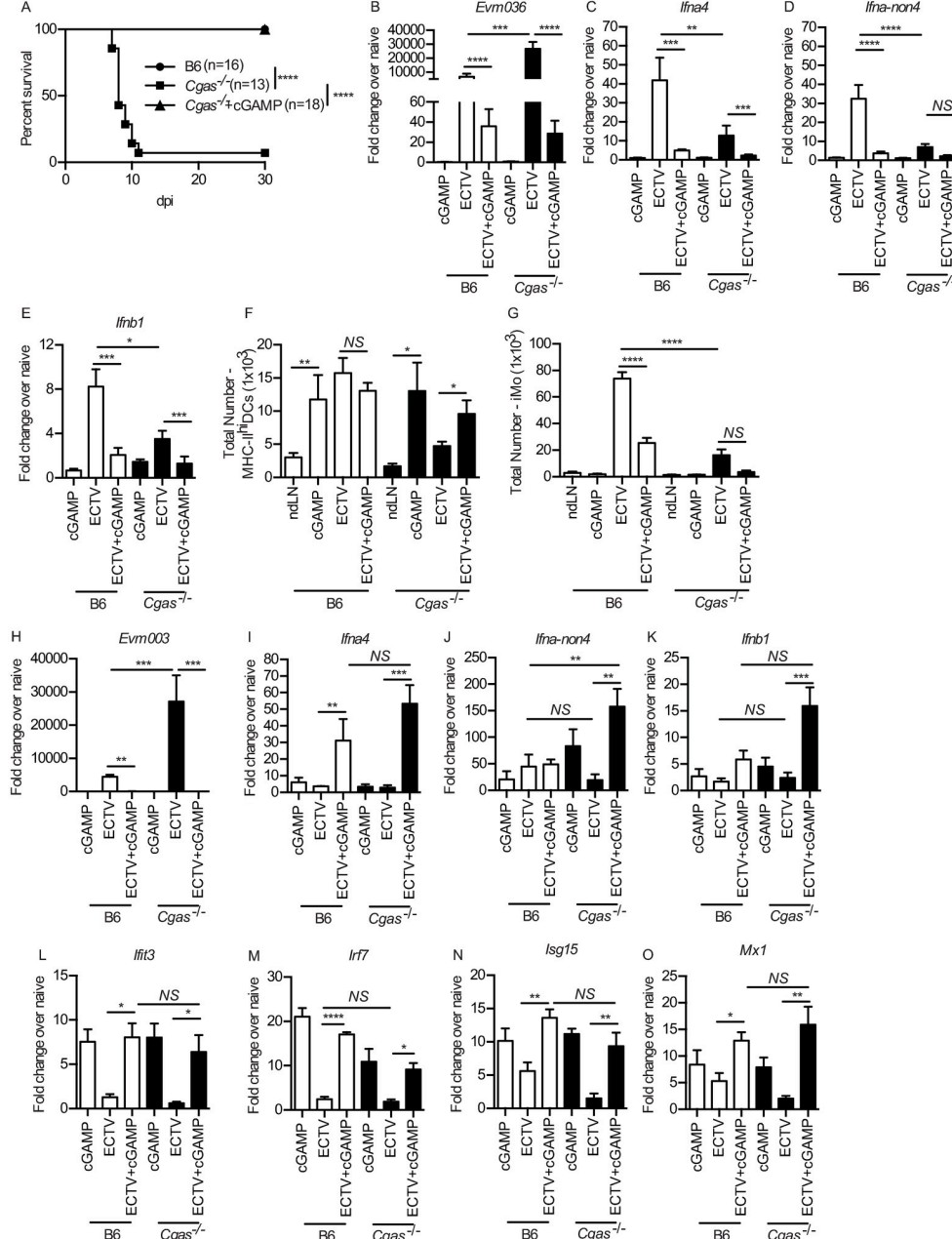

**Fig 5. Administration of cyclic GMP-AMP (cGAMP) to *Cgas*[-/-] mice restores IFN-I expression and resistance to ECTV infection. (A)** Survival of the indicated mice. **(B-E)** Expression of *Evm003* (B), *Ifna4* (C), *Ifna-non4* (D) and *Ifnb1* (E) in the dLN of the indicated mice at 2 dpi as determined by RT-qPCR. Data are displayed as mean ± SEM from 5 mice per group in one experiment, which is representative of three independent experiments. **(F-G)** Calculated total numbers of MHC-II[hi] DCs (E) and iMO (F) in the dLN of the indicated mice at 2 dpi. Data are displayed as mean ± SEM with 12–15 mice per group combined from three similar, independent experiments. **(H-O)** Expression of *Evm003* (H), *Ifna4* (I), *Ifna-non4* (J), *Ifnb1* (K), Ifit3 (L), Irf7 (M), *Isg15* (N) and *Mx1* (O) in the skin of the indicated mice at 2 dpi as determined by RT-qPCR. Data are displayed as mean ± SEM from 5 mice per group in one experiment, which is representative of three similar, independent experiments. For all, *p<0.05, **p<0.01, ***p<0.001, ****p<0.0001.

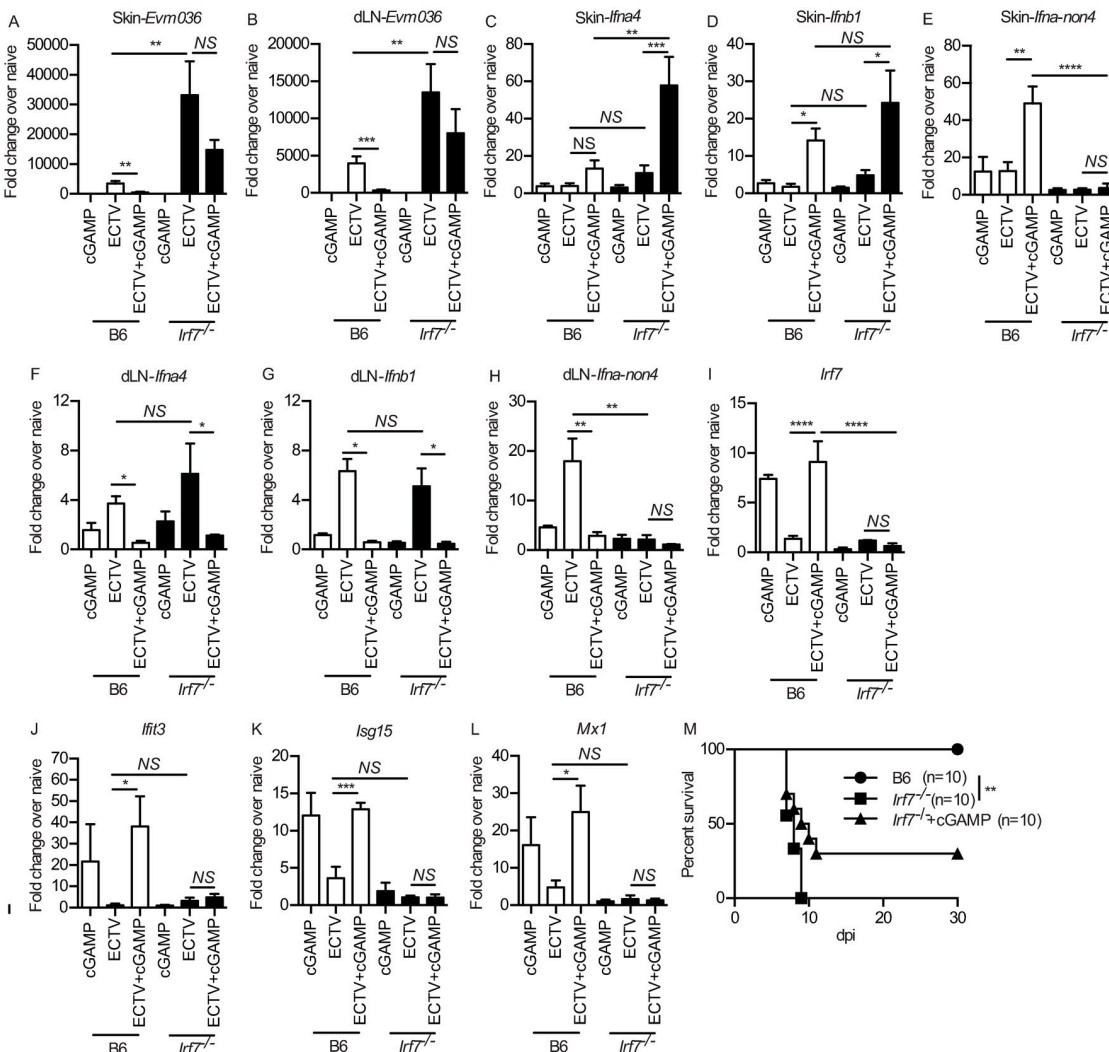

**Fig 6. cGAMP-mediated resistance to ECTV is mostly dependent on IRF7. (A-B)** Expression of *Evm036* in the skin (A) and dLN (B) in the skin of B6 and *Irf7⁻/⁻* mice at 2 dpi as determined by RT-qPCR. Data are displayed as mean ± SEM from 5 mice per group in one experiment, which is representative of three independent experiments. **(C-L)** As in (A-B) but for IFN-I expression in the skin (C-E), dLN (F-H) and ISGs in the skin (I-L). **(M)** Survival of the indicated mice. For all, *p<0.05, **p<0.01, ***p<0.001, ****p<0.0001.

(S5E–S5H Fig) uncoupling IFN-I and proinflammatory cytokine induction. Moreover, cGAMP did not rescue *Ifnar1⁻/⁻* mice from lethal mousepox (Fig 7M). These data indicate a critical role for IFN-I signaling in cGAMP-mediated resistance to ECTV.

## Discussion

Our lab and others have been redefining the paradigm of the LN as not only an important site for lymphocyte priming but as an essential checkpoint to curb viral dissemination. Resistance to mousepox requires a multitude of components of the innate immune system, including a strong NK cell and IFN-I response [16,19,29]. The early accumulation of NK cells in the dLN is required for restriction of ECTV lympho-hematogenous spread and protection from lethal mousepox [19,29]. Moreover, we have recently dissected the pathway in which migratory DCs, group1 innate lymphoid cells (ILCs, mostly NK cells) and iMO collaborate to recruit

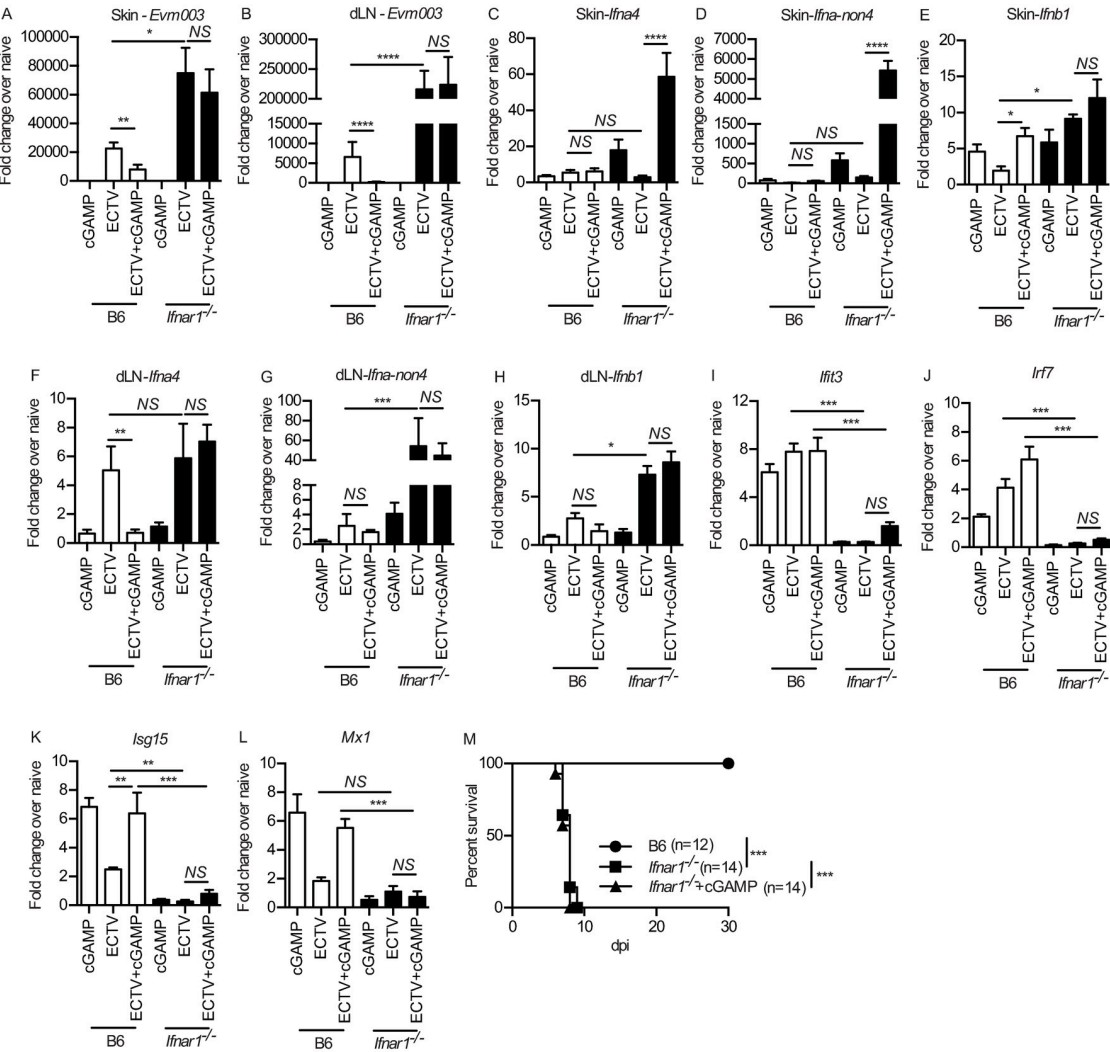

**Fig 7. cGAMP-mediated protection against ECTV is dependent on IFNAR1. (A-B)** Expression of *Evm003* in the skin (A) and dLN (B) of B6 and *Ifnar1*$^{-/-}$ mice at 2 dpi as determined by RT-qPCR. (C-H) Expression of IFN-I in the skin (C-E) and dLN (F-H) of B6 and *Ifnar1*$^{-/-}$ mice at 2 dpi as determined by RT-qPCR. (I-L) Expression of ISGs in the skin of B6 and *Ifnar1*$^{-/-}$ mice at 2 dpi as determined by RT-qPCR. Data are displayed as mean ± SEM from 5 mice per group in one experiment, which is representative of two independent experiments. (M) Survival of the indicated mice. For all, *p<0.05, **p<0.01, ***p<0.001, ****p<0.0001.

circulating NK cells to the dLN [17]. IFN-I exert a wide variety of pleotropic biological effects through the upregulation of ISGs, all of which collectively work to resist and control infection [6]. The importance of IFN-I in acute viral infections is clearly demonstrated through IFNAR1 deficiency or blocking experiments, which predominantly increases viral replication, dissemination and lethality even for many non-species-specific viruses such as Zika virus, West Nile virus or Dengue Virus in mice [8–10,35–38]. After ECTV infection, MHC-II$^{hi}$ DCs migrate to the dLN and are responsible for the very early expression of IFN-I and the chemokines CCL2 and CCL7 that recruit iMO [16]. Notably, TLR9 and MyD88 are indirectly necessary to bring iMO to the dLN and for resistance to mousepox. However, they are not directly necessary for IFN-I production [16]. Instead, once infected, the incoming iMO use STING signaling to IRF7 and NF-κB to induce the expression of IFN-α and IFN-β respectively, thereby becoming the main source of IFN-I in the dLN [16]. However, the mechanisms of IFN-I dependent

resistance and the identity of the PRR upstream of STING that mediate IFN-I induction against acute ECTV infection remained unclear.

Identifying the major cytosolic DNA PRR for IFN-I induction by ECTV is important because ECTV is the archetype for a large number of viruses relevant to animal and human health that utilize the lympho-hematogenous route of dissemination. STING is a critical adaptor protein downstream of over 10 different cytosolic DNA sensors [39]. We recently discounted DAI as the major PRR for ECTV [16]. Another main candidate as an ECTV PRR was IFI204, the murine ortholog for the human IFI16 and member of the AIM2-like receptor family, because it was among the few genes that mapped to Rmp-4 on chromosome 1 that confers resistance to ECTV [22]. Furthermore, IFI204 has been shown to be important in interacting with ASC and procaspase-1 to form the inflammasome and subsequent IL-1β release [24] as well as induction of IFN-β expression in herpesvirus [24], *L. monocytogenes* [40], *F. novicida* [23], and *M. bovis* infection [41]. Our results show, however, that IFI204 is not essential for resistance to mousepox because all *Ifi204*$^{-/-}$ mice survived infection, had similar viral titers in the liver as B6 mice, and their IFN-I expression in the dLN was the same as in B6 mice. Yet, while it appears that IFI204 is not the major PRR for IFN-I induction during ECTV infection, it may still play secondary role, or may have some role in inflammasome activation or a cell-death pathway that contribute to ECTV control. In support, *Ifi204*$^{-/-}$ had slightly higher virus titers in the spleen than B6 mice.

The cGAS-STING pathway mediates IFN-I production *in vitro* in mouse cDCs [42], macrophages and fibroblasts [28], and in human embryonic kidney 293 cells [43] after infection with the OPVs VACV or MVA. Moreover, cGAS-STING-dependent IFN-I production has been implicated in numerous infection models including herpes simplex virus [28,44], adenovirus [45], cytomegalovirus [46], Kaposi's sarcoma-associated herpesvirus [47] as well as retroviruses including HIV-1 and HIV-2 [48]. Yet, the role of cGAS in protection against viruses that disseminate through the natural route of infection in natural hosts had not been studied. Very recently, Cheng and colleagues demonstrated that cGAS-STING-IRF3 is required for optimal induction of IFN-β in various cell types [18]. Furthermore, Cheng et al. showed that cGAS is necessary for increased survival to mousepox at $10^6$ pfu but not to 3,000 pfu ECTV [18]. Our present study partially confirms some of Cheng et al. results and significantly extends them. Different to Cheng et al., we found that *Cgas*$^{-/-}$ mice were highly susceptible to lethal mousepox after infection with only 3,000 pfu ECTV. The reason for the difference between labs is unclear but may be due to differences in animal facilities such as, for example, changes in the microbiome.

Analysis of various transcripts in the dLN revealed that *Cgas*$^{-/-}$ deficiency resulted in drastically reduced expression of IFN-I and downstream ISGs, which is the likely reason why viral replication and spread was increased, Yet, IFN-I and ISG induction was not completely ablated in the dLN of *Cgas*$^{-/-}$ mice. This could be attributed to two reasons: **1)** While cGAS deficiency essentially completely ablates IFN-I expression in iMO and MHC-II$^{hi}$ DCs, there could be other immune cell types that produce IFN-I independently of cGAS. **2)** Other cytosolic DNA sensors may recognize ECTV and induce some IFN-I. While DAI [16] or IFI204 are not required for resistance against ECTV, they may still partially contribute to the IFN-I response. Notably, cGAS deficiency did not affect the transcription of proinflammatory cytokines/chemokines in the dLN indicating that their induction is fully dependent on other sensing pathways, such as TLR9, and that cGAS is likely dedicated to the induction of IFN-I and ISGs. Thus, the need for cGAS reinforces the importance of IFN-I in resistance to mousepox and the notion that cGAS can play a pivotal role in ISG induction after viral infection [49].

The role of cGAS in the hematopoietic vs non-hematopoietic compartment in anti-viral control is unclear. Li et al. demonstrated that *Cgas*$^{-/-}$ mouse fibroblasts failed to produce IFN-I

[28]. Furthermore, cGAS plays a role in human keratinocytes sensing of HSV-1 and VACV [25] and the anti-tumor response in stromal cells [50]. However, we have shown that during ECTV infection, IFN-I is mainly produced by BMD MHC-II[hi] DCs and iMO, suggesting that the critical cells expressing cGAS should be BMD. Using bone marrow chimeras, we show that B6→B6 and B6→Cgas[-/-] mice survived mousepox with significantly lower viral gene expression in the dLN and infectious virus in the liver than B6→Cgas[-/-] and Cgas[-/-]→Cgas[-/-], which succumbed to mousepox. This demonstrates that cGAS in hematopoietic cells but not in other tissues is essential for resistance to mousepox and virus control. Interestingly, B6 → cGas[-/-] had significantly more virus in the liver than B6→B6 mice and Cgas[-/-]→Cgas[-/-] had more virus in the liver than Cgas[-/-]→B6 mice, suggesting a minor contribution in virus control by the parenchymal cells such as hepatocytes, or by radio-resistant, self-sustaining, innate immune cells derived from the yolk sac, such as Langerhans and/or Kupffer cells [51,52]. In support, we have found that Langerhans cells are the main subset of MHC-II[hi] DCs that are responsible for the accumulation of iMO in the dLN [60]. Consistent with the survival data and the results with plain Cgas[-/-] mice, however, IFN-I and ISG induction was significantly higher in B6→B6 and B6→Cgas[-/-] compared to Cgas[-/-]→B6 and Cgas[-/-]→Cgas[-/-] mice, suggesting that cells generated in the bone marrow are the main IFN-I producers and required for ISG expression.

Binding of DNA activates cGAS by inducing a conformational change in the cGAS active site. Activated cGAS catalyzes the synthesis of cyclic GMP-AMP (cGAMP), which functions as a second messenger for IFN-I induction by activating STING [12,28,31]. cGAMP has been used as an adjuvant to boost antigen-specific T cell responses [28] and in cutaneous vaccination against influenza [53], mucosal vaccination [54], and in various tumor models [32,55,56]. We found that local administration of exogenous cGAMP in the footpad bypassed cGAS deficiency and fully rescued Cgas[-/-] mice from lethal mousepox. Mechanistically, cGAMP primed the skin microenvironment to induce IFN-I and ISG transcription after ECTV infection, which reduced the level of local viral replication and, consequently, decreased viral dissemination to the dLN. Thus, the innate responses in the dLN were drastically decreased. Notably, despite that some viral dissemination still occurred, the mice were protected from lethal mousepox. These data indicates that early virus sensing by BMD cells and the subsequent IFN-I response at the initial site of entry may serve as a critical checkpoint to decrease lympho-hematogenous viral dissemination and resist systemic lethal disease. Notably, ECTV infection alone induced less IFN-I mRNA in the skin than ECTV+cGAMP. This could be due to the many immune evasion molecules encoded by ECTV and other poxviruses [57]. For example, VACV inhibits the activation of STING, and Cowpox and ECTV inhibit STING dimerization *in vitro* [58]. Also of interest, cGAMP alone also induced little IFN-I suggesting that in addition to cGAMP production by cGAS, other virus-sensing mechanisms are required for IFN-I expression. Surprisingly, cGAMP administration induced a stronger IFN-I response in the skin of Cgas[-/-] than in B6 mice. Our experiments provide a snapshot of the differential IFN-I response in the skin of B6 and Cgas[-/-] at 2 dpi. While the importance of these differences remain unknown, our data suggest that cGAS may provide a negative feedback signal in the skin, which contains a complex microbiome. Future experiments will focus on the administration of cGAMP at different time points pre- and post-infection, as well as different routes of administration, to assess how cGAMP modulates the innate and adaptive responses and evaluate cGAMP's efficacy as an effective adjuvant in anti-viral therapeutics.

Following cGAMP binding, the endoplasmic-reticulum (ER)-membrane adaptor STING [12] becomes activated and traffics to the ER-Golgi intermediate compartment and the Golgi apparatus, to activate TANK binding kinase 1 (TBK1). It is commonly described that activated TBK1 phosphorylates IRF3 or IKK which then phosphorylates NF-kB [12]. Activated IRF3,

NF-kB and other transcription factors traffic to the nucleus to bind to the promoters and induce the expression of IFN-I and proinflammatory cytokines including IL-6, TNF, and IL-1β [12]. While STING was originally shown to activate not only IRF3 but also IRF7 [33] and IRF3 and IRF7 have been shown to have non-redundant roles in different infection models [59], IRF3 is generally described in reviews as the main IRF downstream of cGAS and STING [28]. Cheng et al. showed that transcription of *Ifnb1* by murine L929 and RAW264.7 cells infected with ECTV required cGAS-STING-IRF3. Yet, they did not confirm their results in IRF3 or IRF7 deficient mice *in vivo* [18]. In previous work we showed that the expression of IFN-α in the dLN is dependent on IRF7 and not IRF3 and that IRF7 but not IRF3 is required for resistance to mousepox [16]. In agreement, here we found that cGAMP did not promote significant survival of *Irf7*$^{-/-}$ mice to mousepox, did not decrease viral replication in the skin, and did not promote *Ifna-non4* expression in the skin which emphasizes that cGAS-STIN-G-IRF7 is the predominant IFN-I-inducing pathway in the skin and dLN during ECTV infection.

Finally, we showed that ECTV+ cGAMP failed to prevent viral replication and lethal mousepox in *Ifnar1*$^{-/-}$ mice. Analysis of gene transcription showed that *Ifnar1*$^{-/-}$ mice upregulated IFN-I and proinflammatory cytokine genes in both the skin and dLN to higher levels than B6 mice probably due to the increased virus loads. Yet, *Ifnar1*$^{-/-}$ mice did not upregulate ISGs. These results indicate that the main mechanism of action of cGAMP is through the induction of ISGs by IFN-I.

In summary, here we demonstrate that cGAS-mediated IFN-I and ISG induction in the hematopoietic compartment is required for protection against ECTV. We also show that increased susceptibility due to cGAS deficiency can be overridden by local cGAMP administration. Mechanistically, cGAMP decreases viral replication in the skin and lympho-hematogenous viral dissemination by activating ISGs *via* a pathway that requires IRF7, IFN-I and IFNAR. Together, these results highlight the importance of cGAS as the major cytosolic DNA sensor to ECTV and provide important insights into the innate mechanism that can curb viral dissemination from the initial site of infection, and its importance for overall virus control.

## Supporting information

**S1 Fig. CpG does not induce IFN-I and ISG expression in the dLN of B6 or *Cgas*$^{-/-}$ mice.** B6 or *Cgas*$^{-/-}$ mice were either infected with 3,000 pfu ECTV or given 25 ug CpG in the footpad. LNs were harvested at 2 dpi. Data are displayed as mean ± SEM from 5 mice per group in experiment, which is representative of two similar experiments. For all, *p<0.05. (TIF)

**S2 Fig. Efficiency of B6-*Cgas*$^{-/-}$ BM chimera reconstitution.** (**A**) Total cellularity of the naïve contralateral LNs of B6-*Cgas*$^{-/-}$ chimeric mice. (**B-G**) Total numbers of the indicated immune cell subtype in the naïve contralateral LNs of B6-*Cgas*$^{-/-}$ chimeric mice. Data are displayed as mean ± SEM with 10–15 mice per group combined from three similar, independent experiments. (TIF)

**S3 Fig. *Cgas*$^{-/-}$ mice have an intrinsic defect in IFN-I expression and accumulation of MHC-II$^{hi}$ DCs, iMOs and NK cells.** (**A**) Frequency of CXCL9$^+$MHC-II$^{hi}$ DCs in the dLN of the indicated mice at 2.5 dpi. (**B-G**) MFI of IFNAR1 (B), costimulatory molecules CD40 (C), CD80 (D) and CD86 (E), and NKG2D ligands Rae1 (G) and MULT1 (G). Data are displayed as mean ± SEM of 10–12 mice per group combined from three independent experiments. (**H-L**) Expression of mRNA for *Ifna-non4* (H), *Ifnb1* (I), *Irf7* (J), *Isg15* (K) and *Mx1* (L) from

sorted uninfected and infected MHC-II$^{hi}$ DCs from the dLN of the indicated mice at 2 dpi. Data are displayed as mean ± SEM of pooled cells from 6–8 mice per group in one experiment, which is representative of two similar experiments. P values were calculated based on three technical replicates. **(M-Q)** As in (H-L) but for iMO. (R-S) Frequency of CXCL9$^+$ and TNF-α$^+$ iMO in the dLN at 2.5 dpi during ECTV infection. **(T-V)** Frequencies of Granzyme B$^+$ (T), IFN-γ$^+$ (U), and TNF-α$^+$ (V) NK cells in the dLN of the indicated mice at 2.5 dpi. For all, *p<0.05, **p<0.01, ***p<0.001, ****p<0.0001.
(TIF)

**S4 Fig. Proinflammatory cytokine and chemokine expression in *Irf7*$^{-/-}$ mice.** (A-H) Expression of proinflammatory cytokines and chemokines in the skin (A-D) and dLN (E-H) of B6 and *Irf7*$^{-/-}$ mice at 2 dpi with or without cGAMP administration. Data are displayed as mean ± SEM from 5 mice per group in one experiment, which is representative of three independent experiments. For all, *p<0.05, **p<0.01, ***p<0.001, ****p<0.0001.
(TIF)

**S5 Fig. Proinflammatory cytokine and chemokine expression in *Ifnar1*$^{-/-}$ mice.** (A-H) Expression of proinflammatory cytokines and chemokines in the skin (A-D) and dLN (E-H) of B6 and *Ifnar*$^{-/-}$ mice at 2 dpi with or without cGAMP administration. Data are displayed as mean ± SEM from 5 mice per group in one experiment, which is representative of three independent experiments. For all, *p<0.05, **p<0.01, ***p<0.001, ****p<0.0001.
(TIF)

## Acknowledgments

We thank Lingjuan Tang for technical assistance.

## Author Contributions

**Conceptualization:** Luis J. Sigal.

**Formal analysis:** Eric B. Wong, Luis J. Sigal.

**Funding acquisition:** Eric B. Wong, Luis J. Sigal.

**Investigation:** Eric B. Wong, Brian Montoya, Maria Ferez, Colby Stotesbury.

**Methodology:** Eric B. Wong.

**Project administration:** Luis J. Sigal.

**Supervision:** Luis J. Sigal.

**Writing – original draft:** Eric B. Wong.

**Writing – review & editing:** Luis J. Sigal.

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
