## [Decision Letter · Decision Letter 0]

11 Sep 2019

Dear Dr. Sigal,

Thank you very much for submitting your manuscript "Resistance to an acute viral disease requires cGAS in bone marrow-derived cells which can be bypassed with cGAMP therapy" (PPATHOGENS-D-19-01493) for review by PLOS Pathogens. Your manuscript was fully evaluated at the editorial level and by independent peer reviewers. The reviewers appreciated the attention to an important problem, but raised some substantial concerns about the manuscript as it currently stands. These issues must be addressed before we would be willing to consider a revised version of your study. We cannot, of course, promise publication at that time. All three reviewers thought your study was technically sound, although concern was raised about novelty because of the recent publication by Cheng et al. Two of the reviewers were more positive, but all three had questions, recommended experiments, and controls. I ask you to address all of the reviewers' comments. In particular, the additional controls recommended by Reviewers 1 and 3 are necessary, as are experiments further characterizing lymphocytes in draining lymph nodes, viral burden in other organs, and assessment of cellularity in lymph nodes after cGAMP + ECTV. Additional experiments to determine whether cGAMP alters cellularity in the lymph nodes of IRF7- and IFNAR1-deficient mice also would be useful. If you choose not to do additional experimentation to address certain reviewer questions, please carefully justify your reasons in your rebuttal. Overall, this is a nice study that will be of interest to readers of PLOS Pathogens with some further experimentation and mechanistic detail. The reviews are appended below.

We therefore ask you to modify the manuscript according to the review recommendations before we can consider your manuscript for acceptance. Your revisions should address the specific points made by each reviewer.

(1) A letter containing a detailed list of your responses to the review comments and a description of the changes you have made in the manuscript. Please note while forming your response, if your article is accepted, you may have the opportunity to make the peer review history publicly available. The record will include editor decision letters (with reviews) and your responses to reviewer comments. If eligible, we will contact you to opt in or out.

(2) Two versions of the manuscript: one with either highlights or tracked changes denoting where the text has been changed; the other a clean version (uploaded as the manuscript file).

Additionally, to enhance the reproducibility of your results, PLOS recommends that you deposit your laboratory protocols in protocols.io, where a protocol can be assigned its own identifier (DOI) such that it can be cited independently in the future. For instructions see http://journals.plos.org/plospathogens/s/submission-guidelines#loc-materials-and-methods

We hope to receive your revised manuscript within 60 days. If you anticipate any delay in its return, we ask that you let us know the expected resubmission date by replying to this email. Revised manuscripts received beyond 60 days may require evaluation and peer review similar to that applied to newly submitted manuscripts.

Sincerely,

Jonathan Miner

Guest Editor

PLOS Pathogens

Michael Diamond

Section Editor

PLOS Pathogens

Kasturi Haldar

Editor-in-Chief

PLOS Pathogens

orcid.org/0000-0001-5065-158X

Grant McFadden

Editor-in-Chief

PLOS Pathogens

orcid.org/0000-0002-2556-3526

Reviewer's Responses to Questions

**Part I - Summary**

Reviewer #1: The authors describe the importance of Cgas in resistance to mouse pathogen ectromelia virus (ECTV). Some of the novelty is taken out by the recent publication by Cheng et al. (as also mentioned by the authors). However, the aim of the study is still important enough for publication by PP as it seeks to deepen the understanding of what cells are central to the phenotype of increased susceptibility in Cgas deficient mice. This contribution is mostly focused in figure 3 and 4. These figures need to increase the number of relevant basic controls for those to be valid. Major and minor concerns are found below:

Reviewer #2: This study by Wong et al described innate immune response to mouse ectromelia virus (ECTV), a poxvirus. The authors performed several sets of in vivo experiments with various mouse knockout strains lacking key components of the DNA sensing and interferon response pathways. They showed that innate immune response to and host protection against ECTV are dependent on cytosolic DNA sensor cGAS, not IFI204, and this is mediated via hematopoietic cells. They also showed that cGAMP treatment in mice confers protection against ECTV through IRF7 and IFNAR.

Reviewer #3: In the manuscript entitled “Resistance to an acute viral disease requires cGAS in bone marrow-derived cells which can be bypassed with cGAMP therapy” Wong et al provide evidence suggesting that cGAS is the upstream sensor of Ectromelia Virus (ECTV) and is required in hematopoietic cells. cGAS deficiency impacts IFN-I expression and accumulation of MHC-IIhi DCs, iMOs and NK cells. cGAMP administration restores resistance to ECTV infection which is dependent on IRF7 and IFNAR1. Even though cGAS has been identified as a sensor during ECTV infection, the authors provide important evidence in terms of the cell types impacted by cGAS deficiency in controlling this virus. Data suggesting administration of cGAMP as a therapy during infections, is also an interesting and promising idea. However, some issues would need to be addressed.

**Part II – Major Issues: Key Experiments Required for Acceptance**

Reviewer #1: Figure 2: Is the difference in IFN-I and ISG dependent on the challenge? What if you challenged with TLR9 agonist CpG? Does the difference persist? This is an important control to test if it is truly an effect of CpG being engaged by the virus – or – a basal lack of IFN-I/ISGs due Cgas deficiency.

Figure 3 is the most important figure in distinguishing this work from the work of Cheng et al. However, some important controls and important analysis are missing:

Fig 3A: How successful was the chimera–generation? Have the authors tested if the adoptively transferred bone.marrow cells led to a functional haematopoetic compartment? Was there a difference in how well the Cgas KO bone-marrow established itself in the recipient compared with the wt bone-marrow? Do the transferred cells have equal amount of TLR9? / or respond equally well to CpG DNA? What about at the time of infection? Has something changed?

Figure 4

Fig 4C, L, and T: How can there be “no difference” in percentage of MHC-IIHI DCs but a clear difference in total number of cells? This must mean that other cell types must also have increased in numbers. How was the number of cells calculated? Was it per weight of the dLN? Or were the dLNs just different in size between mouse strains? May I missed it in the text, but I can´t find a description of this. This should be clarified. Also, the authors should use dots instead of bars to make clear the distribution of data points.

Reviewer #2: I do not have any major concerns regarding the technical aspect of the experiments. Most experiments are very well designed, and the results are convincing (although expected).

However, the conceptual and mechanistic novelty of this study are very limited. Almost all of the conclusions are well expected from existing literature on poxvirus sensing by the cGAS-STING pathway, cGAMP is known to stimulates interferon and antiviral response through IRFs and IFNAR. The Cheng et al study (ref 18) last year used the exact same virus ECTV and has already demonstrated many of the similar observations. I appreciate the extensive efforts the authors put in with many of the in vivo mouse studies presented here, but they reveal very little novel mechanistic insights.

Reviewer #3: Specific issues:

1. In the 50:50 mixed B6-cGAS-/- bone marrow chimeras, the authors show that MHC-IIhi DCs, iMOs and NK cells were the main cell types impacted. Could they also comment on the distribution of other cells in the LN like T cells in these experiments.

2. The authors comment that “cGAMP decreases viral replication in the skin ” however cGAS expression in non-hematopoietic cells has little effect on viral load in the spleen and liver measured at 7dpi (Figure 3). The viral gene expression in the dLN in the same figure is measured at 2dpi. What happens to viral loads in the spleen and liver at 2dpi dose? Is it possible that the role of cGAS in non-hematopoietic cells is dose dependent ? What is the ISG expression in skin in these chimeric mice?

3. The authors mention that the decreased IFN-I expression and increased infection rate of MHC-IIhi DCs and iMOs in the dLN of Cgas-/- mice could increase viral-induced cell death without affecting frequencies of MHC-IIhi DCs and iMOs. Do the authors have a way to quantify this ?

4. Figure 5 points out that administration of cGAMP 4 hrs prior to ECTV infection restores viral resistance however the gene expression data and cellular numbers in LN are not convincing.

“However, there was a drastic reduction in Ifna4 (Figure 5C) Ifna-non4 (Figure 5D) and Ifnb1 (Figure 5E), as well as a significant reduction in total numbers of MHC-IIhi DCs (Figure 5F) and iMOs (Figure 5G).” Please indicate what is the comparison here. Have the authors looked into cell frequencies in LN after cGAMP + ECTV? What about viral load in spleen and liver in ECTV+cGAMP conditions? What cells are infiltrating the skin after cGAMP administration?

5. cGAMP administration to IRF7 KOs does significantly change a set of genes in ECTV+cGAMP condition as compared to ECTV+cGAMP in B6 such as Ifna4 (skin Fig 6C) , Ifna-non4 in skin and dLN (Figure 6 E and H). A set of genes Ifit3, Isg15 and Mx1 were only tested in skin and not in LN. It’s not clear from the data presented if all the effects are IRF7 dependent? Why do a set of genes change and others don’t ? This is observed in other figures too, Figure 5J in B6 mice ECTV compared to ECTV+cGGAMP ifna4 is significantly upregulated but ifna-non4, Ifnb1 are unchanged. cGAMP administration in B6 mice would likely enhance all ISGs.

6. What is the status of MHC-IIhi DCs and iMOs (numbers or frequencies) in dLN in IRF7 or IFNAR deficient mice treated with cGAMP?

7. In Figure 7 C and D why ifna-4 and Ifna-non4 are increased in the skin but not in dLN of IFNAR1 deficient mice , comparing ECTV +cGAMP in B6 to ECTV+ cGAMP in IFNAR KO? Are there different cell types infiltrating these two tissues after cGAMP administration?

**Part III – Minor Issues: Editorial and Data Presentation Modifications**

Reviewer #1: Headline: As the manuscript is only about one single type of virus – then I think “an acute viral disease” should be replaced by the name of the actual disease/infection.

The two separated sections that both refer to figure 1 should be merged.

Fig 4G: Could the authors clarify what this panel measures. Is it uninfected mice that have such a high level of evm003 expression?

Figure 5:

This figure is not really informative besides the point that cGAMP can induce IFN-I, which we already know can protect from infection. Admin of other IFN-I inducing agents, or IFN-I itself, is likely to have same effect. Therefore, these experiments neither support nor stand against the hypothesis that Cgas is important for the resistance.

Figure 6:

From Figure 6M it is clear that the effect of cGAMP is only partly dependent on Irf7. This should be reflected in the figure caption.

Reviewer #2: Some minor comments:

1. cGAS knockout mice has reduced ISG expression at baseline. Some of the ISG measurements (e.g. Figure 2) need to show before and after infection values. The 2-fold decrease in most ISGs after infection may be similar to that of before infection, if so, that will complicate the interpretation of ISGs.

2. Poxvirus antagonize cGAS pathway through poxins that cleaves cGAMP (PMID: 30728498). Does ECTV encode a poxin to antagonize cGAMP?

3. The discussion is too long. Much of the content is comparing the Cheng et al study rather than highlighting the implication of some of the findings.

Reviewer #3: Throughout the manuscript its difficult to follow which conditions are being compared to draw clear conclusions. Additional information in legend or text would help the reader. Please state what conditions are controls and what conditions are being tested.

PLOS authors have the option to publish the peer review history of their article (what does this mean?). If published, this will include your full peer review and any attached files.

Reviewer #1: No

Reviewer #2: No

Reviewer #3: Yes: Katherine Fitzgerald

---

## [Editor Report · Decision Letter 1]

12 Nov 2019

Dear Dr. Sigal:

Thank you very much for submitting your manuscript "Resistance to Ectromelia virus infection requires cGAS in bone marrow-derived cells which can be bypassed with cGAMP therapy" (PPATHOGENS-D-19-01493R1) for review by PLOS Pathogens. Your manuscript was fully evaluated at the editorial level and by independent peer reviewers. The reviewers appreciated the attention to an important topic but identified some aspects of the manuscript that should be improved.

We therefore ask you to modify the manuscript according to the review recommendations (place data in a Supplemental Figure as described below) before we can consider your manuscript for acceptance. 

(1) A letter containing a detailed list of your responses to the review comments and a description of the changes you have made in the manuscript. Please note while forming your response, if your article is accepted, you may have the opportunity to make the peer review history publicly available. The record will include editor decision letters (with reviews) and your responses to reviewer comments. If eligible, we will contact you to opt in or out.

(2) Two versions of the manuscript: one with either highlights or tracked changes denoting where the text has been changed; the other a clean version (uploaded as the manuscript file).

We hope to receive your revised manuscript within 60 days or less. If you anticipate any delay in its return, we ask that you let us know the expected resubmission date by replying to this email.

[LINK]

Sincerely,

Jonathan Miner

Guest Editor

PLOS Pathogens

Michael Diamond

Section Editor

PLOS Pathogens

Kasturi Haldar

Editor-in-Chief

PLOS Pathogens

orcid.org/0000-0001-5065-158X

Grant McFadden

Editor-in-Chief

PLOS Pathogens

orcid.org/0000-0002-2556-3526

The revised manuscript has adequately addressed all of the reviewer concerns. However, prior to acceptance of your revised manuscript for publication, we recommend that some key controls requested by the reviewers are included as supporting data. Specifically, data included in Reviewer Figure 1 and the bone marrow reconstitution efficiency should included in the supporting information, since these results are critical for interpretation of your findings.

---

## [Editor Report · Decision Letter 2]

25 Nov 2019

Dear Dr. Sigal,

We are pleased to inform that your manuscript, "Resistance to Ectromelia virus infection requires cGAS in bone marrow-derived cells which can be bypassed with cGAMP therapy", has been editorially accepted for publication at PLOS Pathogens. 

Before your manuscript can be formally accepted and sent to production, you will need to complete our formatting changes, which you will receive by email within a week. Please note that your manuscript will not be scheduled for publication until you have made the required changes.

IMPORTANT NOTES

(1) Please note, once your paper is accepted, an uncorrected proof of your manuscript will be published online ahead of the final version, unless you’ve already opted out via the online submission form. If, for any reason, you do not want an earlier version of your manuscript published online or are unsure if you have already indicated as such, please let the journal staff know immediately at plospathogens@plos.org.

(2) Copyediting and Proofreading: The corresponding author will receive a typeset proof for review, to ensure errors have not been introduced during production. Please review the PDF proof of your manuscript carefully, as this is the last chance to correct any errors. Please note that major changes, or those which affect the scientific understanding of the work, will likely cause delays to the publication date of your manuscript. 

(3) Appropriate Figure Files: Please remove all name and figure # text from your figure files. Please also take this time to check that your figures are of high resolution, which will improve the readbility of your figures and help expedite your manuscript's publication. Please note that figures must have been originally created at 300dpi or higher. Do not manually increase the resolution of your files. For instructions on how to properly obtain high quality images, please review our Figure Guidelines, with examples at: http://journals.plos.org/plospathogens/s/figures.

(4) Striking Image: Please upload a striking still image to accompany your article if one is available (you can include a new image or an existing one from within your manuscript). Should your paper be accepted, this image will be considered for our monthly issue image and may also appear on our website to feature your article. Please upload this as a separate file, selecting "striking image" as the file type upon upload. Please also include a separate "Other" file with a caption, including credits and any potential copyright information. Please do not include the caption in the main article file. If your image is from someone other than yourself, please ensure that the artist has read and agreed to the terms and conditions of the Creative Commons Attribution License at http://journals.plos.org/plospathogens/s/content-license. Please note that PLOS cannot publish copyrighted images.

(5) Press Release or Related Media: If your institution or institutions have a press office, please notify them about your upcoming paper at this point, to enable them to help maximize its impact. If they will be preparing press materials for this manuscript, please inform our press team in advance at plospathogens@plos.org as soon as possible. We ask that you contact us within one week to plan ahead of our fast Production schedule. If you need to know your paper's publication date for related media purposes, you must coordinate with our press team, and your manuscript will remain under a strict press embargo until the publication date and time. This means an early version of your manuscript will not be published ahead of your final version. 

(6)  PLOS requires an ORCID iD for all corresponding authors on papers submitted after December 6th, 2016. Please ensure that you have an ORCID iD and that it is validated in Editorial Manager.  To do this, go to ‘Update my Information’ (in the upper left-hand corner of the main menu), and click on the Fetch/Validate link next to the ORCID field.  This will take you to the ORCID site and allow you to create a new iD or authenticate a pre-existing iD in Editorial Manager

(7) Update your Profile Information: Now that your manuscript has been provisionally accepted, please log into Editorial Manager and update your profile, if needed. Go to https://www.editorialmanager.com/ppathogens, log in, and click on the "Update My Information" link at the top of the page. Please update your user information to ensure an efficient production and billing process. 

(8) LaTeX users only: Our staff will ask you to upload a TEX file in addition to the PDF before the paper can be sent to typesetting, so please carefully review our Latex Guidelines http://journals.plos.org/plospathogens/s/latex in the meantime.

(9) If you have associated protocols in protocols.io, please ensure that you make them public before publication to guarantee immediate access to the methodological details.

Best regards,

Jonathan Miner

Guest Editor

PLOS Pathogens

Michael Diamond

Section Editor

PLOS Pathogens

Kasturi Haldar

Editor-in-Chief

PLOS Pathogens

orcid.org/0000-0001-5065-158X

Grant McFadden

Editor-in-Chief

PLOS Pathogens

orcid.org/0000-0002-2556-3526
---

## [Editor Report · Acceptance letter]

19 Dec 2019

Dear Dr. Sigal,

We are delighted to inform you that your manuscript, "Resistance to Ectromelia virus infection requires cGAS in bone marrow-derived cells which can be bypassed with cGAMP therapy," has been formally accepted for publication in PLOS Pathogens.

Best regards,

Kasturi Haldar

Editor-in-Chief

PLOS Pathogens

orcid.org/0000-0001-5065-158X

Grant McFadden

Editor-in-Chief

PLOS Pathogens

orcid.org/0000-0002-2556-3526